# IBA Delivery Technique and Media Salts Affected In Vitro Rooting and Acclimatization of Eight *Prunus* Genotypes

**DOI:** 10.3390/plants12020289

**Published:** 2023-01-07

**Authors:** John D. Lawson, William C. Bridges, Jeffrey W. Adelberg

**Affiliations:** 1Department of Plant and Environmental Sciences, Clemson University, Clemson, SC 29634, USA; 2Department of Mathematics, Clemson University, Clemson, SC 29634, USA

**Keywords:** correlative analysis, genotypic variation, micropropagation, multifactor, nutrition, *P. persica*, *P. cerasifera*, *P. munsoniana*, interspecific hybrids

## Abstract

Difficult-to-root plants often perform poorly during acclimatization and in vitro rooting can increase the survival and quality of plants. The influence of auxin application and mineral nutrition on in vitro rooting and subsequent effects on plant quality in eight *Prunus* genotypes were investigated. Microshoots were rooted in vitro on Murashige and Skoog (MS), ½ MS, Driver and Kuniyuki (DKW), or New Prunus Medium (NPM) media formulations in combination with 15 µM indole-3-butyric acid (IBA), 4-day 15 µM IBA pulse, 1 mM 30 s quick-dip, or IBA-free treatments. Shoots were observed pre- and post-acclimatization to determine rooting methods to maximize quality and minimize labor. A genotype-specific response to auxin application was observed with seven of eight genotypes achieving 100% survival when paired with the recommended IBA treatment. Peaches performed best when treated with 4-day IBA pulse or 30 s quick-dip. Rooting of *P. cerasifera*, it’s hybrid to *P. persica,* and *P. munsoniana* all benefitted from IBA application. Shoots rooted with 15 µM IBA were smaller and lower quality in most genotypes. DKW maximized size and quality in six genotypes. Better shoots and larger root systems during in vitro rooting produced better plants in the greenhouse with no detrimental effect of callus growth. Rooting techniques to maximize plant quality while reducing labor are specified.

## 1. Introduction

Fruit tree culture, including *Prunus*, involves planting of orchards that are intended to be cultivated for decades and withstand biotic and abiotic stresses. Newly emerging plant pathogens cause breeders to generate interspecific hybrids to integrate novel alleles into improved germplasm. Interspecific hybrids, often sexually sterile, require vegetative propagation to provide material to tree nurseries. Traditional vegetative propagation of *Prunus* utilizes seasonally available soft and semi-hardwood stem cuttings which slows multiplication and may possibly spread disease. Alternatively, clean, pathogen-free clones can be rapidly multiplied through micropropagation. However, in vitro plant responses to culture medium have been observed to be genotype-dependent and nutrient salt composition needed to be reconsidered to maximize production [1,2]. Commercial laboratories have economized micropropagation by simultaneously rooting and acclimatizing microcuttings via ex vitro rooting. However, in species that are difficult to root directly in a greenhouse, the more conservative in vitro rooting phase may be useful before transfer to a greenhouse environment. 

Genotypic variation in rooting micropropagated *Prunus* has been observed in studies with media salt formulations and plant growth regulators (PGRs) ratios. For almond-peach and peach-almond hybrids, rooting efficiency were higher when treated with a strength of Murashige and Skoog (MS), Driver and Kuniyuki (DKW), and woody plant medium (WPM) relative to standard MS medium with optimal levels of IBA concentrations specific to individual cultivars [3,4,5]. A new Prunus medium (NPM) containing macronutrients similar to ½ MS, and meso nutrients similar to full-strength MS, was shown to be useful in rooting several species of *Prunus* in Oasis^®^ phenolic foam [6]. In peach, a genotype-specific preference in synthetic auxin type and concentration, based on fruit ripening (early, mid, late) was reported [7]. Large, multi-species studies further confirm the need for genotype-specific propagation optimization in *Prunus.* Optimal IBA concentrations ranging from 4–7µM were reported for in vitro rooting of 11 *Prunus* rootstocks but a considerable variation in rooting was observed [8]. Alternative PGR application techniques in *Prunus* induce roots with a short duration of exposure to IBA and followed by transfer to PGR-free medium [5,9]. This method is consistent with reports in *Malus* that showed a short period of auxin exposure followed by the removal of the PGR minimized the inhibition of root elongation and the formation of callus under prolonged IBA exposure [10,11]. However, this two-step method is costly due to the required additional transfer of shoots. A possible alternative to this method is a rapid aseptic application of concentrated IBA during the scheduled subculture, analogous to traditional methods in stem propagation. An aseptic quick-dip has been observed to promote rooting in micro shoots of woody species such as *Malus*, *Pyrus*, *Camellia*, and in other species of *Prunus japonica* Thunb. [12,13,14,15,16].

The present study examined the effect of various inorganic nutrient formulations in combination with methods of IBA application on in vitro rooting of eight *Prunus* genotypes important to ongoing breeding efforts. *P. persica* ‘Guardian^®^’, *P. munsoniana* 59/1, and *P. cerasifera 14/4*, *20-3*, *20-4* serve as potential parental lines in novel rootstock development and *P. persica* × *P. cerasifera* 106.4, *P. persica* ‘GF-305’, *P. persica* × *P. umbellata* ‘MP-29’ were used for their utility in breeding and importance to disease management strategies. Methods were developed to maximize in vitro and greenhouse growth and determine possible correlations between early plant health and subsequent plant size following acclimatization. Additionally, techniques to maximize plant quality while accounting for the time required to apply IBA were considered.

## 2. Results and Discussion

### 2.1. In Vitro Responses

#### 2.1.1. Percent Rooting

The rooting percentage was affected by IBA treatment, genotype, and their interaction (Table 1). A 4-day pulse was most effective with 82–92% rooting, followed by 30 s quick-dip (72–82%), 15 µM IBA constant application (64–75%), and lastly the IBA-free treatment (33–44%). Genotypes clustered in two performance groups based on percent rooting: group one containing both peach genotypes, both hybrids, and plums 14/4 and 20-3 had 65–83% rooting; and group two containing plums 20-4 and 59/1 with 42–61% rooting (Figure 1). The interaction between IBA treatment and genotype was evident. Peach genotypes rooted at a higher rate when paired with either a 4-day pulse, 88–95%, or 15 µM IBA treatments, 82–91%. However, elongated roots were observed in both genotypes in an IBA-free medium following the 4-day pulse treatment. Short, stunted root initials subtended by callus were observed when treated with constant 15 µM IBA (Figure 2). All plums and hybrids performed better when treated with a 30 s quick-dip followed by placement on IBA-free medium compared to the peach genotypes. Both peaches, ‘Guardian^®^’ and ‘GF-305’, had the highest percent rooting when treated with a longer duration of IBA in the 4-day pulse or 15 µM IBA treatments. ‘MP-29’, 106.4, and 20-3, exhibited the highest rooting percentage when treated with either the 4-day pulse (87–97%) or 30 s quick-dip. 59/1 followed this trend with lower rooting, averaging 70–77%. 14/4 and 20-4 performed uniformly across all IBA treatments, with 14/4 averaging 85–87.5% rooting and 20-4 averaging 65–75% rooting. The three myrobalan plums, 14/4, 20-3, and 20-4, and hybrid 106.4 had the lowest rooting in the absence of IBA, indicating the exogenous auxin’s importance. ‘MP-29’ had the same rooting percentage without IBA application and under 15 µM IBA, while 59/1 produced roots at a higher rate when no IBA was applied relative to the 15 µM IBA treatment. ‘GF-305’ rooted similarly when treated with 30 s quick-dip and IBA-free treatments (45–72%). ‘Guardian^®^’ rooted at 34–73% when 30 s quick-dip or when no IBA was administered (Figure 1). A root quality score (data not presented) categorized as the length of roots and the presence or absence of branching, was more indicative of subsequent plant quality after greenhouse acclimatization than the more simplistic percent rooting.

#### 2.1.2. In Vitro Shoot Quality

IBA treatment, genotype, and salt composition affected plantlet quality during in vitro culture (Table 1). Shoot quality reflected the quality of the plant post-rooting with 1 as the maximum score. The 4-day pulse treatment produced shoots with the best quality (0.66–0.78) and shoots treated with 15 µM IBA had the poorest quality (0.23–0.36), with IBA-free and 30 s quick-dip being intermediate of the two (Figure 3). Genotypes were split into three performance groups: group one contained both peaches Guardian^®^ and ‘GF-305’ and 20-3 with a quality range of 0.63–0.79; group two containing both hybrids ‘MP-29’ and 106.4, and plums 20-4 and 14/4 with a quality range of 0.40–0.49, and group three, containing only ‘59/1’ with a quality range of 0.17–0.35 (Figure 4). From all the factors investigated, media salt had the least impact on shoot quality but was still statistically significant. DKW, full MS, and NPM were similar with ranges of 0.47–0.67, whereas plants grown on ½ MS had the poorest quality with a range of 0.33–0.45. ½ MS containing the lowest concentration of all nutrients and the poor-quality shoots indicate insufficient levels to promote high-quality growth.

### 2.2. Correlative Effects

Correlative analysis of in vitro plant responses showed subsequent effects on acclimatization and greenhouse growth with a positive correlation between in vitro quality and plant size (r = 0.57) (Figure 5). The appearance of in vitro quality can be considered as a forward indicator of improved plant growth during acclimatization. Usually, when the shoots had a good appearance in the lab, they would grow larger after transfer to the greenhouse. Likewise, shoots with larger in vitro roots correlated well with larger plants in the greenhouse, (r = 0.47). Longer, branched in vitro roots established better root systems in the greenhouse regardless of possible mechanical damage that could occur during transplant (Figure 6). The presence or absence of callus during rooting treatment did not affect final root quality (r = 0.10) or plant size (r = 0.08) (Figure 5). This was unexpected as prior work has linked callused plantlets with poor ability to thrive in greenhouse and nursery conditions [17].

### 2.3. Ex Vitro Responses

#### 2.3.1. Plant Size

Plant size segregated into three categorical groupings of small, medium, and large plants with corresponding scores of ≤8.7, between 8.7 and 10, and ≥10, respectively (Figure 7). IBA treatment, genotype, salt composition, and their interactions play significant roles in the individual parameters used to construct the size index. The performance of the IBA treatments differentiated greenhouse plants into distinct size groups. In vitro shoots that had been treated with 4-day pulse grew largest with a range of 12–12.9, 30 s quick-dip was intermediate with 10.5–11.3, and the IBA-free and 15 µM IBA were the smallest ranging from 9.3–10.4. This finding is consistent with in vitro observations where plants grown on 15 µM IBA medium had lower quality shoots relative to other IBA treatments. This is potentially linked to reports of prolonged auxin exposure inhibiting root elongation and hindering surface area for nutrient and water exchange [10,18]. This was visually evident in early observations in which plants formed very short, brittle roots subtended by callus (Figure 2).

Plants rooted on DKW were largest with a range of 11.3–12.1, followed by MS and NPM with averages of 10.2–11.1, followed by ½ MS with a range of 9.7–10.5. The length of the longest leaf, a factor in the plant size, was highly influenced by salt formulation and drove this response in the index indicating a higher leaf length when treated with DKW. DKW and ½ MS produced plants with the best developed root system but ½ MS had the smallest leaves and stems. Early growth provided by high nutrient concentrations from DKW prepared the plantlets treated with this salt for the transition to autotrophic growth in the mist bed environment. In traditional *Prunus* propagation by stem cuttings, leaf area has been documented to influence rooting where a greater canopy results in better rooting [19,20]. Plantlets with a larger canopy were able to push new leaf growth that in turn promotes a growing root system and results in high-quality plants (Figure 8) [19]. An interaction between genotype and IBA treatment was observed (Table 1). The peaches ‘Guardian^®^’ and ‘GF-305’ performed best when treated with 4-day pulse. Hybrids ‘MP-29’ and 106.4, and plums 20-4, 20-3, and 59/1 were largest with 4-day pulse and were only distinct only from 15 µM IBA. 14/4 performed similarly across all treatments (Figure 9). Salt interaction with IBA treatment was also significant for plant size. 4-day pulse produced the largest plants across all salt combinations while IBA-free resulted in the smallest.

Replicate was significant in two *ex vitro* responses, number of leaves and length of the longest leaf, that were used in the index construction of plant size (Table 1). Replicates were grown asynchronously in the same greenhouse environment during the same season but were likely affected by changes in sunlight. The shoot canopy varied based upon the mineral formulation in sterile culture, so plants were better apt to capture light and grow more rapidly in later replicates with brighter sunlight. Subsequent effects of media composition used in sterile culture effected canopy size and greenhouse growth in turmeric with the effects of *in vitro* mineral nutrition becoming evident during acclimatization [21]. Likewise, reports of varying responses in vitro and ex vitro growth based upon mineral nutrition has also been described in *Lemna minor* and *Landoltia punctata* [22].

#### 2.3.2. Plant Survival following Acclimatization

Survival of 100% following in vitro rooting was observed when genotypes were paired with the most efficient in vitro rooting treatment, except for Guardian^®^ which survived at 92% (Table 2). 94% of shoots that developed roots before acclimatization survived transfer to the greenhouse growth compared to 85% survival when unrooted shoots were transplanted. However, 73% of plantlets transplanted with roots exhibited growth, a vigor score of 1 or greater, in the greenhouse after a two-week period while only 29% of the unrooted shoots exhibited growth during ex vitro rooting and acclimatization (Figure 10). This finding is consistent with reports of higher growth and acclimatization associated with shoots rooted in vitro in other woody species. Teak was found to have a greater plant height and survival in shoots rooted in culture relative to shoots rooted ex vitro after 7 weeks with a significant effect of the number of roots present [23]. Likewise, shoots of other fruit trees like *Pyrus*, *Malus*, and *Prunus avium*, acclimatized better when rooted in vitro when compared to the transfer of unrooted shoots to a greenhouse environment [24,25].

### 2.4. Best Practices

In most tissue culture systems, plants remain in a fixed concentration of PGR’s during rooting. Part of this study was to consider separating root induction and root growth by temporal IBA-treatments and document the extra work for a root induction treatment during the 4-day pulse and 30 s quick-dip. Labor cost is approximately 60% of micropropagation costs and percent rooting must reach approximately 80% to be economically viable [26,27]. The in vitro IBA treatments were divided into component tasks to provide insight into the labor-based effort for each IBA treatment type for in vitro rooting (Table 3). Preparation of media and vessels required approximately 30 s per plant for IBA-free, 15 µM IBA, and 30 s quick-dip treatments. The 4-day pulse required an additional set of vessels to be made and therefore requires 60 s per plant in media preparation. The subculturing of plants from multiplication medium to IBA-free and 15 µM treatments needed no additional handling and therefore required approximately 30 s of hood time to transfer each microshoot (or 60 s per microshoot when media preparation is considered). The quick-dip consisted of 30 s partial immersion for each microshoot and was done in batches of five to improve efficiency so the treatment of each microshoot using the quick-dip method required 38 s (and 68 s when media preparation was included). Four-day pulse treatment required approximately 30 s per microshoot to cut and place on 15 µM IBA media. However, the additional transfer after four days to IBA-free media required an additional 27 s per microshoot. When the preparing two unique media is considered, the 4-day pulse required 117 s to treat each microshoot.

Balancing effort and plant quality (labor with survival and size) were considered in recommended micropropagation practices for each genotype (Table 2). If treatments were similar, the treatment with the lowest labor requirement was recommended. Maximum ex vitro rooting and largest plant size coalesced into a single recommended treatment for any of the genotypes. 100% survival was observed in all plants within the recommended treatment combination that maximized plant quality and minimized labor with the sole exception of ‘Guardian^®^’ which had a maximum survival of 92%.

The largest plant (20-3) grew the best with MS medium, the smallest plant 59-1 grew best with NPM, and the other 6 genotypes grew best when grown on DKW medium. MS has the largest number of macronutrients. DKW also has almost as many macronutrients, ample mesonutrients, but much larger amounts of calcium and sulfate, than the other media and was most often the optimal media. NPM has much less macronutrients, similar to ½ MS, but the same mesonutrients as MS, and which was recommended for the genotype that produced the smallest plant size when treatment conditions were maximized. ½ MS provides the least amount of mesonutrients, calcium, magnesium, sulfur, and micronutrient manganese, with the low concentration of ammonium, potassium, and nitrate. In vitro plant quality in our trials requires adequate meso-nutrients as is consistent with published multifactor media optimization research in *Pyrus* and *Prunus* [28,29,30]. Our trial of varied complete salt formulations with 4 media factors lacks the resolution of these other) investigations that identified the main nutrient factors and their possible interactions with other components.

## 3. Materials and Methods

### 3.1. Establishment 

Aseptic shoot cultures were established following previously described methods [6]. Briefly, dormant *P. persica* ‘GF-305’ and *P. persica × P. umbellata* ‘MP-29’ shoots were washed in dilute liquid detergent, followed by a 1 min dip in 70% ethanol and subsequent rinse in deionized water, followed by immersion in 10% bleach for 10 min. Shoot buds were peeled and placed in Baxter shell vials containing 20 mL Murashige and Skoog (MS) [18] adjusted with 1 N NaOH to a pH of 6.2. All other genotypes were established via seed stratification and sterilization. Fruit exocarp was sterilized by soaking in 20% bleach followed by 70% ethanol soak, 10 min each, then moved to a laminar flow hood. Ovules were removed from the fruit, placed in vials containing 10 mL Woody Plant Medium and stratified at 4 °C in the dark for 10 weeks. Germination was induced and seedlings were transitioned to maintenance and multiplication medium via aseptic micropropagation by cultivating shoot tips on 0.7% agar medium in Magenta GA/7 (Magenta Corp., Chicago, IL, USA) vessels and grown in a laboratory clean room. Ambient growing conditions were maintained at 22 ± 1 °C and plants were grown under fluorescent lighting at 40 µmol s^−1^ m^−2^ with a 16 h photoperiod.

### 3.2. In Vitro Stock Culture

Culture methods were selected to maintain plant quality during recurrent subculturing and were tailored independently to each genotype. *P. persica* ‘Guardian^®^’ and ‘GF-305’ were maintained consistently on 40 mL of modified Quoirin and Lepoivre (QL) [31] solidified with 0.7% agar supplemented with 30 g L^−1^ fructose, 1 mg L^−1^ ferulic acid, 2 mg L^−1^ ascorbic acid, 100 mg L^−1^ inositol, 2 mg L^−1^ glycine, 1 mg L^−1^ thiamine, 1 mg L^−1^ nicotinic acid, 1 mg L^−1^ pyridoxine, 2.2 µM 6-benzylaminopurine (BA), 0.025 µM indole-3-butyric acid (IBA), and 0.15 µM gibberellic acid (GA_3_) as described in Kalinina and Brown [9]. Plantlets were multiplied by shoot tip culture and transferred every 4–6 weeks. The other *Prunus* genotypes: *P. cerasifera* 14/4, 20-3, 20-4, *P. munsoniana* 59/1, *P. persica × P. cerasifera* 106.4 and *P. persica × P. umbellata* ‘MP-29’ were maintained using New *Prunus* Medium (NPM) [6] supplemented with 1 mg L^−1^ ferulic acid, 2 mg L^−1^ ascorbic acid, 100 mg L^−1^ inositol, 2 mg L^−1^ glycine, 1 mg L^−1^ thiamine, 1 mg L^−1^ nicotinic acid, 1 mg L^−1^ pyridoxine, with cycling between multiplication medium (2.2 µM BA) and resting medium (16 µM IAA). Media cycling was done to maintain healthy cultures and avoid prolonged exposure to BA. All media were autoclaved prior to use.

### 3.3. In Vitro Rooting 

Root induction was observed using a full-factorial arrangement of four media salt components and four IBA application techniques for the eight genotypes. Media containing salt formulations NPM [6], Murashige and Skoog (MS) [32], ½ MS, and Driver and Kuniyuki (DKW) [33] were prepared using a common organic supplement of: 15 g L^−1^ sucrose, 1 mg L^−1^ ferulic acid, 2 mg L^−1^ ascorbic acid, 100 mg L^−1^ inositol, 2 mg L^−1^ glycine, 1 mg L^−1^ thiamine, 0.5 mg L^−1^ nicotinic acid, and 0.5 mg L^−1^ pyridoxine as described by Kalinina and Brown [11]. Media were then divided based on plant growth regulator IBA treatment. One treatment used IBA-free media. The second treatment followed the 4-day pulse of Kalinina and Brown [11]. Briefly, shoot tips were cut and moved to 20 mL aliquots of root induction medium containing half-strength MS and 15 µM IBA in Magenta GA/7 (Magenta Corp., Chicago, IL, USA) vessels for 4 days and then moved to vessels containing IBA-free media. The third treatment used medium supplemented with 15 µM IBA. The fourth treatment received a 30 s immersion in an aseptic 1 mM aqueous K-IBA (Product ID:I560; Phytotech Laboratories, Shawnee Mission, KS, USA) solution, followed by transfer to the respective IBA-free medium. Two hundred milliliters of 1 mM K-IBA solution was dispensed into an RV-750 vessel (EightomegaFive, Santa Paula, CA, USA) with an inert polycarbonate tray (Osmotek LLC, Rehovot, Israel) to keep the shoots upright and prevent total submersion, then autoclaved. Plantlets were treated in batches of five for 30 s then removed from the solution and moved to their respective treatment vessels containing 150 mL IBA-free media. 

The in vitro rooting experiment was conducted in two runs: run 1 contained the seedlings of ‘Guardian^®^’ and ‘GF-305’. Seven ‘Guardian^®^’ seedling clones were observed as independent genotypes but pooled during analysis due to the similarities in responses. Run 2 contained the myrobalan plums 14/4, 20-3, 20-4, goose plum—59/1, and peach-plum hybrids ‘MP-29’ and 106.4. Each vessel contained 15 plants, five plants (experimental units) of each of the three genotypes and was conducted in two technical replicates. Vessels were grown at an ambient temperature of 22 ± 2 °C under fluorescent lighting at 50 µmol s^−1^ m^−2^ with a 16 h photoperiod. After two weeks, the microporous vents integrated into the lid of the vessels were opened. Run 1 were grown for two additional weeks under a vented environment while the plums in run 2 were grown for one additional week. Plantlets were given a quality score based on the appearance of the shoot system using the following criteria: −1—chlorotic or necrotic tissue, no notable growth; 0—no notable growth, green tissue present; 1—green tissue present, notable leaf expansion, and growth. The presence or absence of callus was recorded, the number of roots was counted, and root quality was scored as follows: 0—no roots; 1—roots present, <5 mm in length; 2—roots present and elongated > 5 mm; 3—roots present, elongated, and with secondary roots present.

### 3.4. Acclimatization and Greenhouse Growth

Plantlets and unrooted shoots were gently removed from the agar matrix and then rinsed in tap water to remove residual media. Unrooted shoots were dipped in a 1 mM K-IBA solution for 5 s prior to transplanting. Treatment runs were transferred into polypropylene seedling trays composed of 63 cells measuring 1.5” × 1.5” × 3.5” (Proptek LLC, Watsonville, CA, USA) and filled with Fafard 3B soilless mix (SunGro Horticulture, Pendleton, SC, USA). The acclimatization phase was conducted from November to March in a greenhouse environment at 34.68° latitude with shade cloth and under a 3 mm polyethylene tent. Peak light intensity was approximately 150 µmol s^−1^ m^−2^ at bench height and maintained at 75% ± 5 relative humidity by micro-mist irrigation emitters. The growing area was maintained at 25 ± 2 °C during peak sunlight and 20 ± 2 °C night temperature. Dry cells were hand watered as needed. Individual growth was quantified following the conclusion of the two-week growth period to include- stem height (mm) measured from the soil surface to the shoot tip, the number of leaves > 1 cm, and the length (mm) of the longest leaf. Root quality was scored after removing each plant from its cell and using the following scale: 0—no roots present; 1—roots present in the upper half of cell; 2—roots present throughout the cell; 3—roots present throughout the cell and soilless media plug remained intact by root adhesion after removal from the tray. A greenhouse vigor grade was assigned by observing the aerial tissue of each plant and using the following criteria: −1—chlorotic or necrotic tissue present, no notable growth; 0—green tissue present, no notable growth; 1—green tissue present, notable leaf expansion, and new leaves; 2—green tissue present, notable leaf expansion, new leaves, and elongated internodes. Ex vitro survival was calculated as a percent of plants that did and did not root in vitro and received a greenhouse vigor grade other than −1. Greenhouse growth was calculated as a percent of plants that received a greenhouse vigor grade of 1 or 2 for rooted or unrooted shoots from in vitro culture. Plant size was quantified by an index value using measurements in the following formula:(1)Plant Size=(Mean Stem Height×0.2)+(Mean Length of Longest Leaf×0.2)+(Mean Number of Leaves×0.2)+(Mean Final Root Quality×0.4)

### 3.5. Experimental Design and Statistical Analysis 

In vitro rooting was conducted using a randomized block design. Eight *Prunus* genotypes were screened using a combination of treatment factors of four nutrient salt formulations, four IBA application treatments and replicated asynchronously. Treatment factor combinations were averaged to develop mean responses for each value. Data were analyzed using a 4 × 4 × 8 full factorial model with the inclusion of replicate as a blocking factor using JMP 16.1 (SAS Inst., Cary, NC, USA). An analysis of variance (ANOVA) was used to determine if the primary model terms effects with factors considered significant. *p* < 0.01 was considered evidence of statistical significance. Performance groups of responses were created using 95% confidence intervals. Multivariate correlations were used to quantify relationships between laboratory and greenhouse responses.

## 4. Conclusions

While propagation techniques within *Prunus* vary broadly by genotype, trends in recommended practices based on genotypic lineages exist. Genotypes with peach lineage performed best when treated with the 4-day pulse. Genotypes with myrobalan plum lineage were not sensitive to prolonged exposure to IBA but also responded well to a 30 s quick-dip. The 30 s quick-dip, an alternative rooting method to the 4-day pulse, requires less time and resulted in similar-sized plants in three of the eight genotypes screened. Little prior research has been done on quick-dip applications of exogenous PGRs under sterile conditions and improvements in this technique may warrant further investigation. The 4-day pulse treatment usually worked well but required the most technician time. Mineral nutrition during in vitro root induction was important and significantly affected final plant quality with DKW conferring the best quality in vitro with 6 of 8 genotypes screened. Replicate was significant in some ex vitro response potentially from the asynchronous replication and sunlight quality associated with the replicate. This study provides in vitro rooting methods that were useful for the different *Prunus* genotypes most used in our rootstock disease resistance breeding program.

## Figures and Tables

**Figure 1 plants-12-00289-f001:**
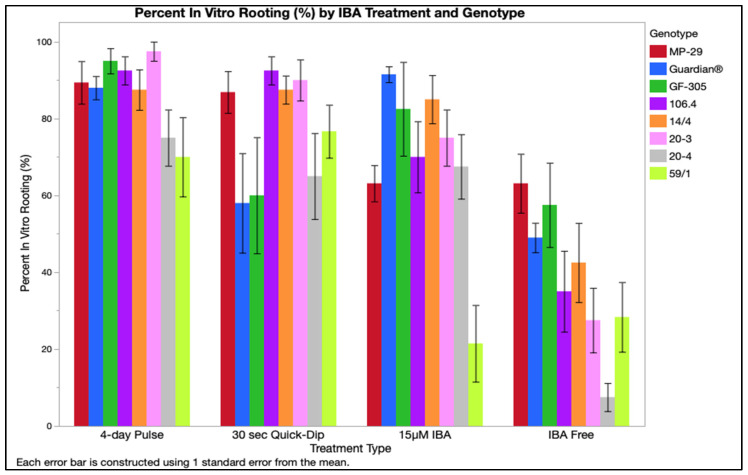
Percent in vitro rooting by genotype and treatment type averaged for four nutrients with two replicates accompanied by standard error bars.

**Figure 2 plants-12-00289-f002:**
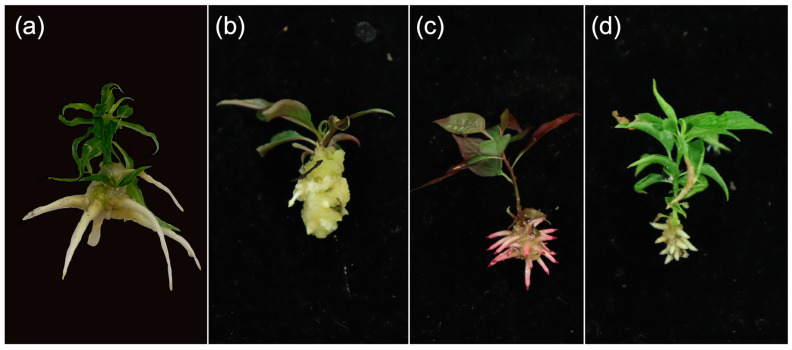
Root and callus formation on microshoots treated with 15 µM IBA. ‘Guardian^®^’ (**a**), ‘MP-29’ (**b**), myrobalan plum 20-4 (**c**), and peach-plum hybrid 106.6 (**d**).

**Figure 3 plants-12-00289-f003:**
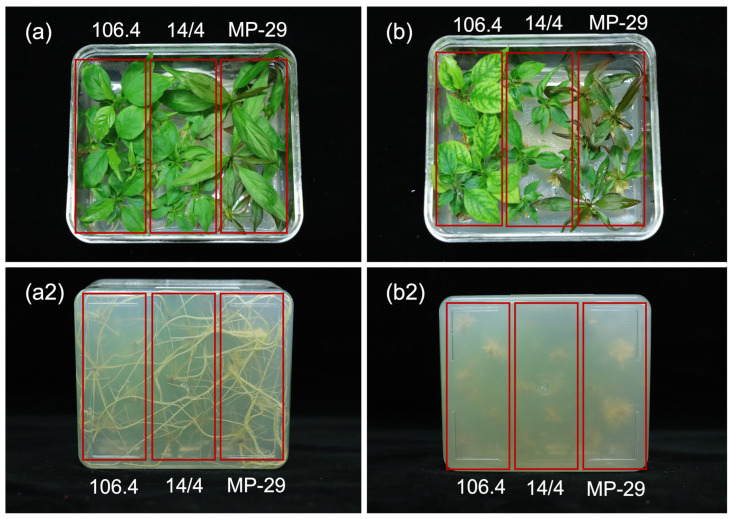
Differences in plant canopy and root system of plum genotypes when treated with a 4-day pulse (**a**,**a2**), and 15µM (**b**,**b2**) IBA treatments on DKW media. Genotypes are arranged vertically and denoted by boxes: left to right, *P. cerasifera × P. persica* 106.4, *P. cerasifera* 14/4, *P. persica × P. umbellata* ‘MP-29’.

**Figure 4 plants-12-00289-f004:**
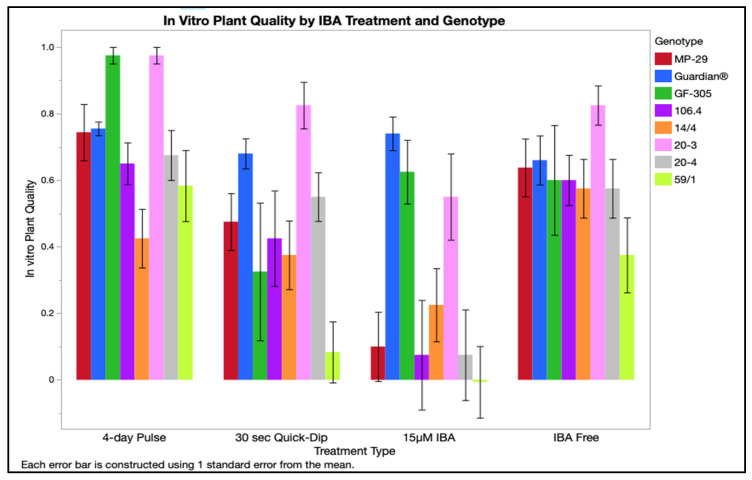
In vitro plant quality by genotype and treatment type was scored using the following: −1—chlorotic or necrotic tissue; 0—no notable growth, green tissue present; 1—green tissue present, notable leaf expansion and growth. Treatments were averaged for four nutrients with two replicates accompanied by standard error bars.

**Figure 5 plants-12-00289-f005:**
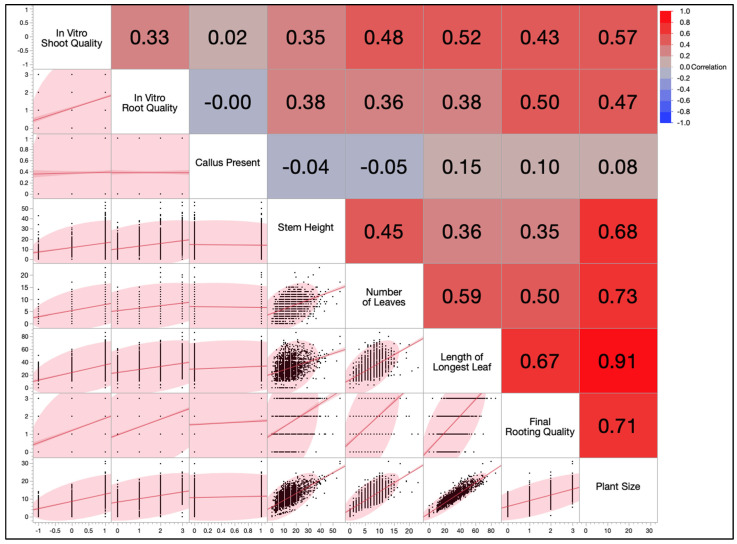
Pearson’s correlation values of in vitro and ex vitro responses across eight *Prunus* genotypes treated with varied media salt (MS, 1/2MS, DKW, NPM) and IBA applications (IBA-free, 15 µM IBA, 30 s quick-dip, 4-day pulse) showed early plant metrics correlate with later plant qualities indicating a subsequent effect of in vitro conditions. *Df =* 1873.

**Figure 6 plants-12-00289-f006:**
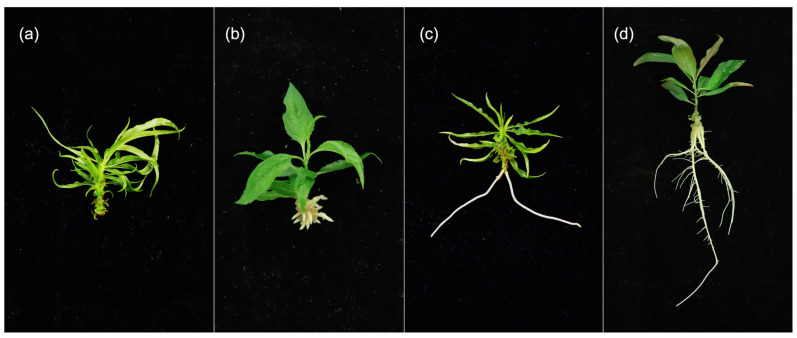
Representative plants displaying the categorical rating of in vitro shoots’ root quality on a scale from 0–3: 0—no roots are present (**a**), 1—roots > 1 cm present (**b**), 2—roots present and elongated (**c**), 3—roots present, elongated, and possess secondary branching (**d**).

**Figure 7 plants-12-00289-f007:**
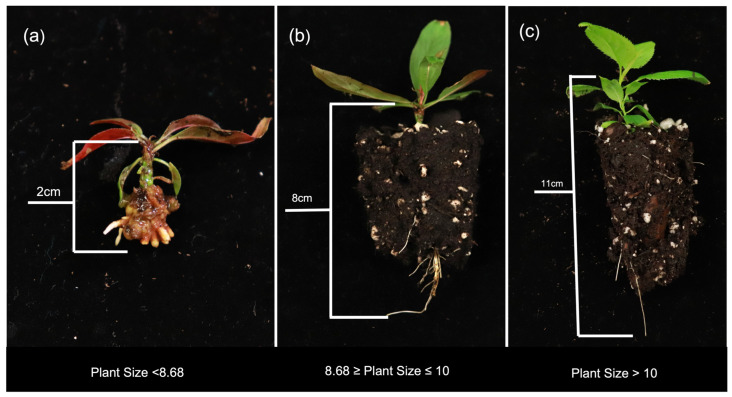
Representative plants displaying the differences seen across plant size calculations of three categories: small (**a**), medium (**b**), and large (**c**) size plants.

**Figure 8 plants-12-00289-f008:**
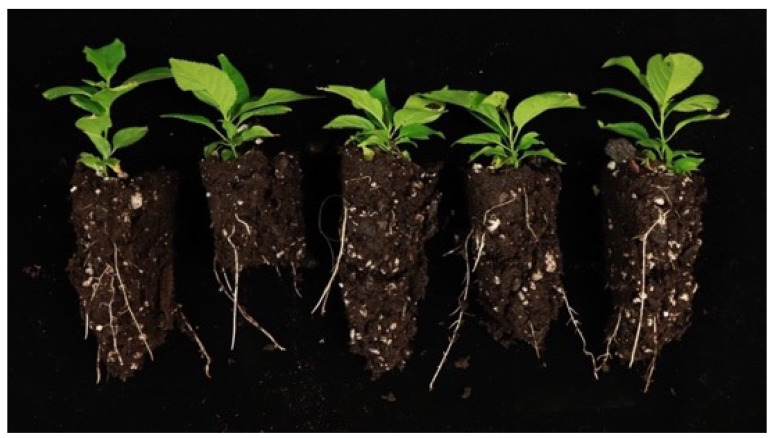
Hybrid 106.4 (*Prunus persica* × *P. cerasifera*) treated with 4-day pulse-DKW combination at the end of the five-week study. Plants developed large leaf canopies and root systems occupying most of the cell.

**Figure 9 plants-12-00289-f009:**
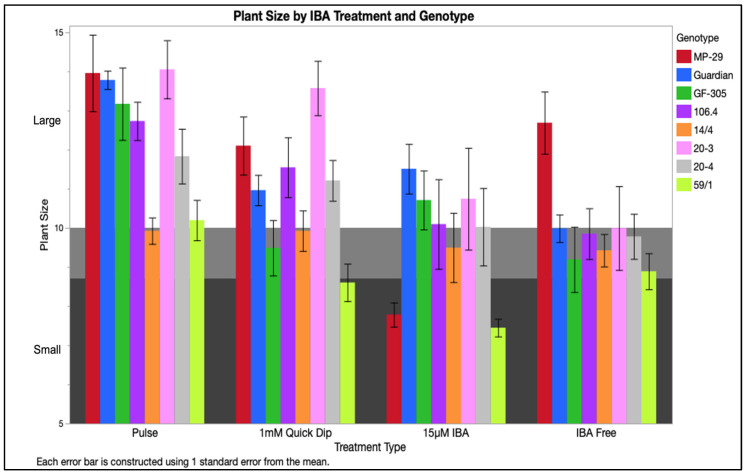
Plant size by IBA treatment and genotype was quantified using an index factoring stem height, number of leaves > 1 cm, length of the longest leaf, and a rooting quality score. Small (<8.86), medium (8.68 ≥ Plant Size ≤ 10), and large plant (>10) ranges indicated by background color. Treatments were averaged for four nutrients with two replicates accompanied by standard error bars.

**Figure 10 plants-12-00289-f010:**
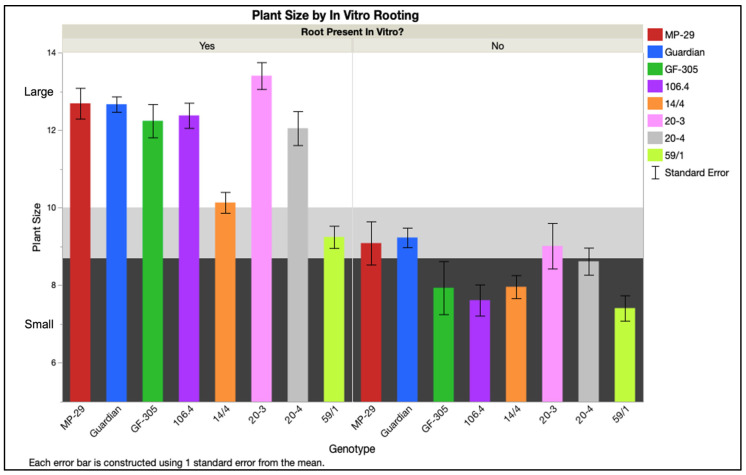
Plant size for each genotype was affected by the presence or absence of roots formed in culture prior to acclimatization. Small, medium, and large plant size ranges indicated by background color. Average responses for four IBA treatments, four nutrients, and two replicates are accompanied by standard error bars.

**Table 1 plants-12-00289-t001:** Summary ANOVA of in vitro responses percent rooting and shoot quality, and ex vitro responses plant size, length of the longest leaf, the number of leaves, stem height, and final rooting quality in response to treatment factors nutrient salt, IBA treatment, genotype, and replicate in eight *Prunus* genotypes.

		In Vitro Responses				Ex Vitro Responses									
Effect		% Rooting		Shoot Quality		Plant Size		Length of the Longest Leaf		Number of Leaves		Stem Height		Final Rooting Quality	
	df	F-Ratio	*p* Value	F-Ratio	*p* Value	F-Ratio	*p* Value	F-Ratio	*p* Value	F-Ratio	*p* Value	F-Ratio	*p* Value	F-Ratio	*p* Value
Salt (S)	3	NS	0.0662	8.85	<0.0001	10.9	<0.0001	19.6	<0.0001	4.67	0.0040	NS	0.6453	9.9	<0.0001
IBA Treatment (T)	3	60.6	<0.0001	33.4	<0.0001	36.0	<0.0001	26.7	<0.0001	20.0	<0.0001	23.8	<0.0001	NS	0.3056
Genotype (G)	7	7.70	<0.0001	15.4	<0.0001	14.0	<0.0001	41.6	<0.0001	3.45	0.0021	22.6	<0.0001	14.6	<0.0001
S × T	9	NS	0.1377	NS	0.0161	4.3	<0.0001	4.72	<0.0001	3.08	0.0022	NS	0.1182	NS	0.6350
S × G	21	NS	0.2683	NS	0.3232	NS	0.7838	NS	0.7026	NS	0.7848	NS	0.8461	NS	0.7669
T × G	21	3.83	<0.0001	2.70	0.0004	3.8	<0.0001	4.65	<0.0001	2.11	0.0060	NS	0.0065	NS	0.0531
S × T × G	63	NS	0.3185	NS	0.1111	NS	0.1837	NS	0.0938	NS	0.7242	NS	0.9121	NS	0.9998
Replicate	1	NS	0.0293	NS	0.0549	43.2	<0.0001	57.9	<0.0001	46.8	<0.0001	NS	0.0428	NS	0.5796
Model Fit (R^2^)		0.77		0.77		0.79		0.85		0.69		0.74		0.63	

Note. Terms were considered significant at *p* = 0.01. NS signifies term was not significant.

**Table 2 plants-12-00289-t002:** Recommendations for salt-IBA treatment combinations to achieve the highest percent acclimatization survival and plant size in eight *Prunus* genotypes.

	Nutrient Salt	Treatment Type	% Survival	Plant Size
Genotype				
*P. persica × P. umbellata* ‘MP-29’	DKW	Quick Dip	100	13.9
*P. persica* ‘Guardian^®^’	DKW	4-day Pulse	92	13.9
*P. persica* ‘GF-305’	DKW	4-day Pulse	100	13.5
*P. persica × P. cerasifera* 106.4	DKW	Quick Dip	100	13.6
*P. cerasifera* 14/4	DKW	15 µM IBA	100	13.1
*P. cerasifera* 20-3	Full MS	Quick Dip	100	14.7
*P. cerasifera* 20-4	DKW	15 µM IBA	100	13.5
*P. munsoniana* 59/1	NPM	4-day Pulse	100	11.7

Survival was calculated as a percent of plants that did and did not root in vitro and received a greenhouse vigor grade other than −1 after 14 days post-transfer. Plant size was quantified by an index value using measurements length of the longest leaf, number of leaves, stem height, and rooting quality.

**Table 3 plants-12-00289-t003:** Time requirements for four IBA delivery techniques used for in vitro rooting given on a per-shoot basis with consideration given to media preparation and cleaning.

	IBA-Free	4-Day Pulse	30 s Quick-Dip	15 µM IBA
	Time Requirements (s Per Shoot)			
Media preparation and dishwashing	30	60	30	30
Cutting and Sticking	30	30	38 *	30
Additional transfer	0	27	0	0
Total time required	60	117	68	60

* Quick dip was conducted during the transfer and accrued 8 additional seconds of labor per microshoot.

## Data Availability

The data presented in this study are available on request from the corresponding author.

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
