# Peer review of "IBA Delivery Technique and Media Salts Affected In Vitro Rooting and Acclimatization of Eight Prunus Genotypes"

_plants, 2023, doi:10.3390/plants12020289_

Round 1

Reviewer 1 Report

Comments concerning manuscript plants-2107168

This research represents an integrated study of in vitro cultivation and subsequent acclimatization of difficult-to-root Prunus species. By comparing the morphological characteristics among differently treated Prunus varieties, the authors have identified in vitro rooting techniques and mineral requirements that maximize plant quality at the acclimatization stage. In line with findings that plantlet quality during in vitro rooting stage is crucial for the successful plant growth ex vitro, screening for the optimal, genotype-specific auxin application method and mineral nutrition with high rooting potential and subsequent growth, and with the least labor cost, would be of substantial economic importance.

The authors provided sufficient background and included all relevant references. The research design is appropriate and the methods used are described adequately. However, the results are not presented clearly, in order to support the conclusions. The authors need to respond to suggested corrections and comments, included in the attached pdf file. 

For the clarity purposes, particularly for the wide readership, I would strongly suggest the authors to designate a more common name to each genotype/variety’s latin name, as they did in their paper “In vitro co-culture system for Prunus spp. and Armillaria mellea in phenolic foam rooting matric” (Adelberg et al. 2021), such as peach rootstock, ‘GF 305’ or peach × plum hybrid, ‘MP-29’, etc. This was done non-systematically and only sporadically in the present manuscript, e.g. Both peaches, ‘Guardian®’ and ‘GF-305’ (page 2, lines 84-85). Different varieties should preferably be introduced in the form of Table, in the Material and Methods section, under the Plant Material subsection. This would make it so much easier for the reader to focus on the authors’ actual findings, instead of trying to decode sentences that use ‘common names’ while at the same time referring to the figures that only contain numerical descriptions of the genotypes.

Reviewer 2 Report

Dear Authors,

Your work is  well presented, interesting.

A very few points to consider appear in the pdf  manuscript.

Reviewer 3 Report

General comments:

The research is well described, conducted, and written. The subject is of interest to many researchers working on woody plant TC. However, the authors need to address few points:

The authors need to give sufficient details of the species and genotypes used in the study and why were these selected. I found the reason in the last sentence of the manuscript! Please indicate this at the end of introduction and in more detail in Materials and Methods. I suggest adding a subheading on “genotypes” in M&M because your title says eight genotypes, so readers will be interested in these.

However, my main concern is the design of the experiment. Why seedlings of two registered cultivars – Guardian and GF 305 were used instead of the clone? This needs to be explained. Also, there is confusion with regards to maintaining the cultures of these (which says “…recurrent subculturing and were tailored independently to each genotype. P. persica ‘Guardian®’ and ‘GF-305’ were maintained consistently on 40 mL of modified Quoirin and Lepoivre (QL) [17] solidified with 0.7% agar…”), with the rooting method (which says, “The in vitro rooting experiment was conducted in two blocks: block 1 contained the seedlings of ‘Guardian®’ and ‘GF-305’. Seven ‘Guardian®’ seedling clones were observed as independent genotypes.”). You maintain Guardian in TC and use seedling progenies for experiment??Why Guardian had seven seedlings? What about GF 305? How many seedlings and why were seedlings used? You use one clone for other genotypes and eight for Guardian. How was this replicated and managed?  In statistics we need some clarity. 

Finally, how can the recommendations derived from results for Guardian seedlings be valid for Guardian clone? Same question for GF 305.

Correlation coefficients need to be tested for significance and indicated in Fig 5. How many degrees of freedom? Please change associated text about correlations after testing the significance and reporting it. The font size is too small to read with the background colour in this Figure.

Finally, the authors have decided to combine results with discussion, but there is hardly any discussion of the results. A bit more discussion (with reference to other published literature) around key results would add value to the manuscript. Changing the proliferation stage cytokinin to meta-Topolin helps with difficult-to-root genotypes ( https://doi.org/10.1007/s11240-019-01597-4 ). This has been tested in Prunus also ( https://doi.org/10.1007/s11240-014-0489-1 ). Some discussion around the cytokinin and subsequent rooting is welcome, so that the readers are given an option of testing your protocols with material propagated in different cytokinins. Maybe, you can plan a future experiment with meta-Topolin and discuss that aspect!

Specific comments:

Title page – Full addresses needed. Title is appropriate.

Abstract – all abbreviations should be explained first, including PGR and media names.

What is the recommended IBA treatment (Line 16)?

268 – There are no ovules in seed! You have embryos if these are fertilised seed.

Keywords – Give species names used in the study rather than very generic words like correlative analysis, multifactor and genotypic variation. Propagation also should change to micropropagation.

Materials

The title is about Prunus. Prunus includes fruit species cherries (Prunus pensylvanica), apricots (Prunus armeniaca), peaches (Prunus persica), plums (Prunus domestica), nectarines (Prunus persica), and almonds (Prunus dulcis). Other ornamentals also exist. Authors should therefore name the species they worked at the outset (end of introduction and beginning of Materials & Methods) without making reader to scroll through to find what these ‘difficult-to-root’ species they worked on. Also give some indication in title and/or Keywords. In line 64 they use words ‘eight Prunus varieties’, thus further confusing the reader. In lines 314 onward, they talk about two blocks, one with two Guardians and another with goose plum, myrabalan plum and hybrids. Also, why these eight? What significance these have commercially? Not all readers are aware of these ‘special’ genotypes.

Results

LINE 102 – P in italics

Line 104 – remove word ‘were’

110-111 – There are two qualities in table – shoot and rooting. So please specify. IBA treatment is not significant for root quality.

Fig. 3 – Need to mention that the plantlets are in vertical rows and there are three genotypes. With poor quality photo, it was difficult to understand where these plantlets are. 

119 - .change to  “…..but was still statistically significant.”  If the media type had least impact, then why ½ MS had the poorest quality in all genotypes?

Section 2.2 – Why use word ‘subsequent’ in subtitle? What you have calculated is the correlations between in vitro parameters and ex vitro parameters and also between different in vitro parameters, for example between in vitro root quality and in vitro shoot quality (r=0.33). Symbol for correlation is r, not R. This table is not complete. Which of these correlations are significant?

Figure 6 has not been referred to in text. Have I missed it?

170-172 Rewrite sentence, unclear and erroneous.

174 – Does DKW has higher mineral concentrations than MS?

175 – Why do you call DKW a salt? It is a growth medium. This was used throughout the text, suggest changing to medium – the widely used term. When used in plural (salts), it sounded OK, but here it sounds odd. In line 171, affected by media salt composition would be better?

180 – IBA treatment type?

Sentence in 185-86 – Merge with previous paragraph. Not good to have single sentence paragraphs.

Figure 9. Specify this as ex vitro plant size in title

 Subtitle 2.3.2 Suggest changing to “Plant survival following acclimatization”. Percentage is the parameter used to test it.

204-206 – So, 85% unrooted plantlets survived in greenhouse but only 29% exhibited growth. Reader is confused here (if M&M is not checked). Therefore, please mention “after two weeks of greenhouse growth” – I am getting this from line 341.  Also refer to the criteria used to assess (not in detail).

 245 what do you mean by “preferred by the smallest plant”? Please rewrite.

247 – Table 4 doesn’t provide the information given in text. Please remove it, I mean the reference to the table. Maybe you can refer to a paper/papers by the authors of media that gives this info or just delete reference to table.

251 – DoE is not used anywhere, please remove.

Table 4 – Shouldn’t the title be in plural? Recommendations for combinations – there are several combinations. Survival % and Plant size measurements need some explanation in a foot note or in title. Tables and figures should stand on their own without having to refer too much to text. Survival after how many days? How was size measured etc.?

313-315 – Why were seedlings used for registered clones? The results and recommendations may be not valid for the clone itself!

I like to see a comment on the reason for statistical significance within replicates for some parameters (Table 1) – could this be due to asynchronous replication (line 359)?

331 – I guess K-IBA solution is a commercial preparation – please give source.

373 – No need to refer to tables in conclusion

380 – This final sentence finally gives a clue to why these genotypes were used in the study. This should be stated upfront with more detail and justification why this should be published in an international journal. All readers are not breeders, or even Prunus researchers, but all of them want to know why.

Round 2

Reviewer 3 Report

Thank you for carefully addressing all the comments. The manuscript is now acceptable for publication.